# Revealing principles of autonomous thermal soaring in windy conditions using vulture-inspired deep reinforcement-learning

Yoav Flato[1,2,3], Roi Harel [4,5,6,7], Aviv Tamar [8], Ran Nathan [2,7] &
Tsevi Beatus [1,2,3] ✉

Thermal soaring, a technique used by birds and gliders to utilize updrafts of hot air, is an appealing model-problem for studying motion control and how it is learned by animals and engineered autonomous systems. Thermal soaring has rich dynamics and nontrivial constraints, yet it uses few control parameters and is becoming experimentally accessible. Following recent developments in applying reinforcement learning methods for training deep neural-network (deep-RL) models to soar autonomously both in simulation and real gliders, here we develop a simulation-based deep-RL system to study the learning process of thermal soaring. We find that this process has learning bottlenecks, we define a new efficiency metric and use it to characterize learning robustness, we compare the learned policy to data from soaring vultures, and find that the neurons of the trained network divide into function clusters that evolve during learning. These results pose thermal soaring as a rich yet tractable model-problem for the learning of motion control.

Vultures and other large birds soar by using thermal updrafts, which are localized warm air columns that rise due to their buoyancy. Thermal soaring enables these birds, as well as human glider pilots, to fly for long distances with high energetic efficiency and under challenging wind conditions[1–5]. These impressive capabilities make bird-like thermal soaring a model for imitation in unmanned aerial vehicles (UAVs)[6]. More broadly, thermal soaring is an appealing model-problem for studying motion control, and how motion control is learned by animals and by autonomous engineered systems. The main reasons are that thermal soaring has rich dynamics and nontrivial constraints, yet it uses relatively few control parameters compared, for example, with cursorial animals and robots. Additionally, thermal soaring is relatively simple to model owing to the lack of contact forces and the availability of reliable and simple quasi-steady aerodynamic force models[7]. Finally,

thermal soaring is becoming more experimentally accessible: on the one hand, data on the flight kinematics of soaring birds are becoming available[1] and, in parallel, implementing thermal soaring in small-scale radio-controlled gliders with onboard AI-based control has become affordable at a relatively low budget[8–10].

Indeed, recent pioneering studies have achieved autonomous thermal soaring in simulated environments and in unmanned gliders using various methods. Rule-based algorithms use state-estimators for the position of the thermal's center and the glider's energetic state, and rely on a set pre-programmed rules to achieve thermal soaring[11–21]. Although these algorithms are pragmatic, effective, and thoroughly comprehended, they do not explore innovative solutions and, therefore, have constraints in advancing our understanding of motion control as a learning challenge.

[1]Rachel and Selim Benin School of Computer Science and Engineering, The Hebrew University of Jerusalem, Jerusalem 9190401, Israel. [2]Department of Ecology, Evolution, and Behavior, Alexander Silberman Institute of Life Sciences, The Hebrew University of Jerusalem, Jerusalem 9190401, Israel. [3]Grass Center of Bioengineering, The Hebrew University of Jerusalem, Jerusalem 9190401, Israel. [4]Department for the Ecology of Animal Societies, Max Planck Institute of Animal Behavior, Konstanz 78467, Germany. [5]Department of Biology, University of Konstanz, Konstanz 78457, Germany. [6]Centre for the Advanced Study of Collective Behaviour, University of Konstanz, Konstanz 78457, Germany. [7]Movement Ecology Lab, Department of Ecology, Evolution, and Behavior, Alexander Silberman Institute of Life Sciences, The Hebrew University of Jerusalem, Jerusalem 9190401, Israel. [8]Department of Electrical and Computer Engineering, Technion, Haifa 3200003, Israel. ✉e-mail: tsevi.beatus@mail.huji.ac.il

This limitation can be alleviated by using machine learning (ML) and, particularly, reinforcement learning (RL) methods[22–25]. In RL, an agent operates in an environment to maximize a specific goal function, referred to as a reward. Reward maximization is done via a learning process and, as the agent learns, it acquires a policy that determines its behavior. The policy controls the agent's actions based on a set of available observations of the environment, which is known as the agent's state. This setting enables the agent to explore the environment and discover its own strategy. Thus, in an ideal learning process, the agent can find an optimal policy whose complexity is limited only by the agent's computational architecture. The agent's policy may even exceed the performance of policies based on human intuition or expertise[26].

The first applications of RL to thermal soaring were presented by Wharington et al. in a simulation[27] and by Reddy et al. in a flying glider, where they trained a rule-based RL agent using the SARSA algorithm operating in an environment free of significant horizontal winds[8,28]. Their agent was represented by a lookup table, which defined the glider's discretized action (bank angle) as a function of its discretized current state (vertical velocity and bank angle). This system achieved successful thermal soaring both in simulations and on a real glider. Although the policy of rule-based RL can be easily interpreted, it is likely that richer ML architectures may achieve better gliding performance[29], especially under more challenging environments, for example, with thermals that are drifted by horizontal winds. This particular condition is probably the norm rather than the exception in thermal soaring[1,4].

Currently, the most relevant ML architecture for such tasks is deep neural networks (NN). Combined with state-of-the-art RL algorithms (deep-RL), such as actor–critic and policy gradient[30–32], these methods have led to numerous successful applications in complicated motion control and planning tasks[22–25,29,33]. In particular, deep-RL methods have been successfully implemented in autonomous thermal soaring both in simulated environments and, more impressively, in real gliders. Novati et al. showed that deep-RL can robustly solve a gliding control problem of a simulated elliptical object, achieving gliding and landing in a complex flow environment[34]. Notter et al. implemented thermal soaring both in simulation and in a real glider by using deep-RL with a long short-term memory (LSTM) network architecture, which is a recurrent-NN which implements memory of previous states and actions[9,10]. Moreover, Notter et al. used deep-RL to treat the thermal soaring problems in two levels: first, locating and exploiting an individual thermal under horizontal wind conditions[10] and, second, solving the decision-making strategy in cross-country soaring. Such strategy should balance two, often conflicting objectives: locally, it should exploit thermal updrafts to gain altitude and, globally, it should race between designated way-points[9,35]. The exploration-exploitation problem in thermal soaring was also addressed by Cui et al., who solved it in a simulated environment using deep-RL and combining new energy considerations[36].

The success of deep-RL models in handling thermal soaring opens the way for using such models to address basic questions related to the learning of motion control. These include, for example: what is the structure of the problem in terms of its bottlenecks – sub-problems that must be solved sequentially to achieve the final behavioral goal[37]? What sensors and actuators are crucial for successful learning? How robust is the acquired policy? Can we dissect the agent's NN – a computational object that is typically treated as a "black box" – to gain insight into its function[38–40]? And, how does a learned policy compare with the thermal soaring behavior of birds in the wild?

Here, we address these questions by developing and studying a simulation-based deep-RL solution for thermal soaring under challenging horizontal winds. Analyzing the reward dynamics revealed that this problem has at least two bottlenecks that can be alleviated by specific modulations of the reward function. The robustness of the RL

policy and of the learning process was characterized by evaluating the agent's performance in unencountered environmental conditions and under noise, as well as by comparing various state and action representations and different learning dynamics. We show, for example, that curriculum learning, in which the horizontal wind speed gradually increases during the learning process is crucial for achieving a policy that can handle different wind speeds. Sensing the ambient wind and controlling the angle-of-attack are also crucial. Additionally, we analyzed the activation patterns of the agent's NN "brain" and revealed distinct neuronal patterns associated with specific behavioral modes that evolve during the learning process, i.e., with the agent's "age". And, finally, we identified similarities between the learned policy and the soaring technique and learning dynamics observed in free-ranging vultures. Overall, we believe that these results both contribute for improving deep-RL implementations in thermal soaring, and more broadly, pose thermal soaring as a rich yet tractable model-problem for complex motion control.

## Results
### Thermal soaring under horizontal wind

Using deep-RL, we trained a NN-based agent in a simulated environment to locate and soar in thermal updraft (Fig. 1 and Methods section). The environment included one thermal that drifted horizontally at speed $u$ (Fig. 1c, d and Methods section). The chosen parameters of the system, referred to as the "nominal" configuration, are as follows. The agent's state at a given timestep consisted of six parameters $\{V, v_z, \sigma, \alpha, \theta, u\}$ (Fig. 1b, e, f), with overall ground speed $V$, climbing rate $v_z$, bank angle $\sigma$, angle-of-attack $\alpha$, angle with respect to the wind $\theta$, and wind speed $u$. Additionally, the state included a memory buffer of these parameters for the 8 previous 1 s time steps. The agent's action was $\{\Delta\sigma, \Delta\alpha\}$, which are the controlled modulations in $\sigma$ and $\alpha$, respectively. The reward per each simulation step was a sum of two components: the agent's climb rate $v_z$ and a penalty $P_{center}$ equal to $-d/50$, where $d$ is the agent's horizontal distance from the thermal center in meters. The thermal center is not part of the agent's state. Assigning such penalty is only feasible for training in simulations, as ground truth for the updraft center is hardly available in a real-world scenario. Yet, training with this penalty may still be indirectly advantageous in these cases (see Discussion). Additionally, the agent was penalized in two cases. First, if the agent went unstable, defined as spinning 360° in pitch, the simulation was stopped and the agent was penalized by $P_{stab}$, equal to −1 per each second left until the end of the run. Second, if the agent crashed ($z = 0$), the simulation was stopped and the agent was penalized by −1000. We used the deep deterministic policy gradient (DDPG) algorithm where both actor and critic NNs included two fully connected hidden layers with 200 neurons each (Methods and Supplementary Note 3). Sample thermalling trajectory is shown in Fig. 1g.

We found that using curriculum learning was crucial for achieving successful soaring under horizontal winds. To this end, we gradually increased $u$ during the learning process. First, the agent was learning on environments with $u$ randomly selected within the range 0–2 m/s. Then, the range of $u$ was increased to 0–4 m/s and, finally, to 0–6 m/s. Each of these three steps consisted of 3·5·10⁶ s of simulated flight.

Agents with the nominal configuration achieved stable thermal soaring under horizontal winds of $u = 0–6 m/s$ (Fig. 2 and Supplementary Movies 1, 2). Figure 2a shows representative trajectories of the same agent, soaring under $u = 1$, 3 and 5 m/s. As agents were, on average, placed initially close to, but not within, the thermal, the trajectories include a short searching phase of 5–20 s that consisted of a circling motion. Once the agent located the thermal, it started thermalling: circling the thermal center and simultaneously tracking it downwind. To track the drifting thermal, the agent decreased its turning radius every time its trajectory is aligned with the wind, i.e., at angle of $\theta \approx \pm180°$ with respect to the wind (in the +$x$ direction). This motion is most evident in the $u = 5 m/s$ trajectory, where the trajectory

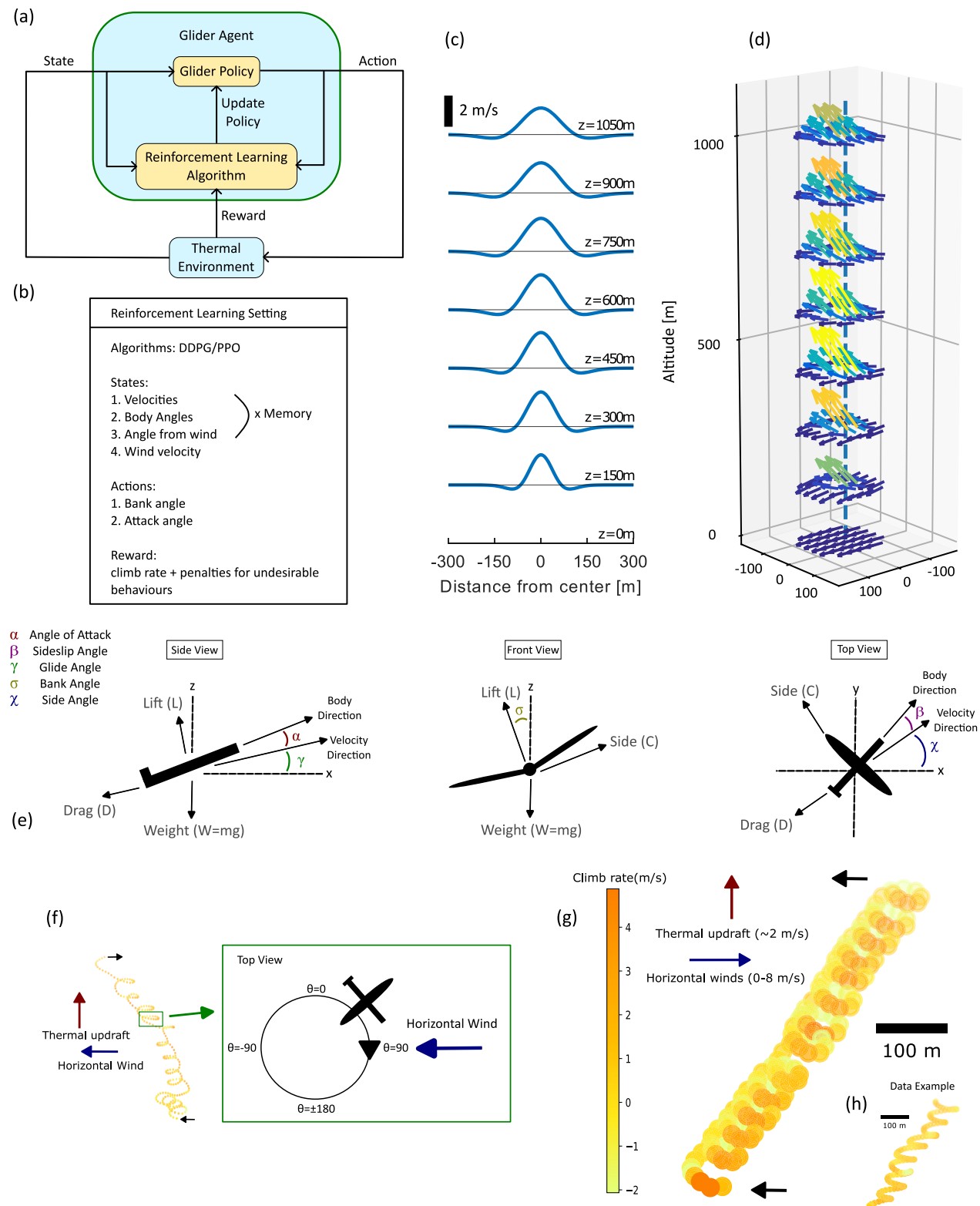

becomes almost parallel to the $x$ axis. The resulting trajectories are trochoid in the $xy$ plane, which manifests a thermalling technique similar to the techniques of soaring vultures[1] and professional human gliders[4,5]. Each learning session resulted in an agent that consistently rotated either clockwise or anti clockwise; the agent in Fig. 2 is of the clockwise type. During thermalling, the agent keeps itself in a range of 10–50 m from the thermal center, with an average distance of ~25 m (Fig. 2b–g). The agent's climb rate $v_z$ consists of two contributions: the updraft velocity of the thermal (peak velocity $\sim 2\,m/s$), and the

horizontal wind velocity, which the agent learned to convert to vertical velocity when flying against the wind.

To understand the agent's thermalling policy, we tested the agent on 100 random initial conditions for each of the three horizontal wind values $u = 1, 3$, and $5\,m/s$. Figure 2e–g shows two-dimensional (2D) histograms of the agent's position with respect to the moving thermal center, calculated for each $u$ across all 100 runs. The histograms show that the agent learned to move around the thermal center in a radius of ~25 m. Interestingly, for $u \geq 3\,m/s$, the center of the circling motion

**Fig. 1 | Reinforcement learning system for simulation-based thermal soaring.**
**a** Overall system layout. **b** The state, action, reward, and algorithm settings of our
RL system. **c** Thermal updraft distribution as a function of distance from the
thermal center $r$, for different heights. The thermal model parameters are $w^* = 5$,
$z^* = 2000$ (Simulation Model in Methods section). **d** The velocity field of a thermal
with horizontal wind. Each arrow represents wind velocity direction and magnitude
at a given $(x, y, z)$ position. Arrows are color coded by the magnitude of $v_z$. The
horizontal wind is $u = 3 \, m/s$ and thermal parameters identical to the thermal in (**c**).
**e** The glider was modeled as a point mass with lift force $L$, drag force $D$, side force $C$,
and weight $W$. The three views show the angles that define the glider's orientation:
the angle-of-attack $\alpha$; the sideslip angle $\beta$ between the body axis and the velocity;
the glide angle $\gamma$; the bank angle $\sigma$ and the side angle $\chi$ of the velocity with respect to
the horizon. **f** The angle from wind, $\theta$ during circling motion. Left: a trajectory of a

thermalling vulture taken from data[1]. Right: a scheme of one circle within the
thermalling motion that demonstrates the values of $\theta$ (the angle between the glider
velocity and the wind $u$ in the $xy$ plane) along the circle. $\theta = 0$ represents motion
against the wind (headwind) and $\theta = \pm180°$ represents motion in the wind direction
(tailwind). $\theta$ is defines such that it is increasing during the circling. Here, for
example, the glider is rotating clockwise, hence when it goes from headwind to
tailwind, $\theta$ increases from 0 to 180°. For counter clockwise rotations, $\theta$ is mirrored
such that it still increases when transitioning from headwind to tailwind. **g** A
representative flight trajectory of the nominal agent configuration, which shows
climbing thermalling motion under horizontal wind (blue arrow). The agent's climb
rate is color coded along the trajectory. **h** A representative flight trajectory of a free-
ranging vulture during thermal soaring[1]. Scale bars are 100 m.

does not align with the thermal center, but the peaks of the 2D dis-
tributions show that along its circular trajectory, the agent spends
more time closer to the thermal center. In addition, in this part of the
trajectory, the agent chases the moving thermal ($\theta \approx \pm180°$, tailwind).
Analyzing $v_z$ across the trajectories, shows that the agent gains most
of its vertical velocity at angles of $\theta \approx 90°$ with respect to the
wind (Fig. 2h–j). For $|\theta| < 90°$ the agent moves against the wind, and for
$|\theta| > 90°$ it moves along the wind. For $u = 5 \, m/s$, we see a clear signature
of the agent's vertical acceleration as it rotates in $\theta$, through the slope
of $v_z(\theta)$. The vertical acceleration is highest between $\theta = -45°$ and $90°$
when the agent mostly flies against the wind, showing it exploits the
wind to increase $v_z$. As $u$ increased, the variability of the agent's $\sigma$ and $\alpha$
increased, indicating a higher control load. For example, the standard
deviation of $\sigma$ increased from $6°$ in $u = 0$ to $12°$ in $u = 5 \, m/s$.

To quantify the agent's soaring performance, we first calculate the
fraction of its flight time spent in the thermal, i.e., $80 \, m$ from its center,
where the thermal updraft is still positive. Figure 3a shows that under
horizontal winds of $0 - 6 \, m/s$ the agent spends most of its flight time
thermalling. To quantify soaring performance in terms of the agent's
mean climb rate $v_z^{agent}$, we define a soaring efficiency metric, $\eta$, which
uses two limits for the climb rate: an upper bound $v_z^{optimal}$ and a
baseline $v_z^{baseline}$:

$$\eta = \frac{v_z^{agent} - v_z^{baseline}}{v_z^{optimal} - v_z^{baseline}}. \tag{1}$$

The upper bound, $v_z^{optimal}$, was estimated for a thermal with the para-
meters used in our environment: $w^* = 5 \, m/s$ and $z^* = 2000 \, m$ (Simula-
tion Model in Methods section and Eq. (3)). It was derived from the
equations of motion (Eq. (2)) by characterizing their steady state under
constant ($\sigma, \alpha$) in the wind frame-of-reference. The steady state radius
was then used to calculate the $v_z$ upper bound in a circular trajectory
within the thermal. Under these settings, the upper bound for the
climb rate is $v_z^{optimal} = 0.72 \, m/s$ (see Supplementary Note 2 for the full
derivation). The second limit of the climb rate $v_z^{baseline} = -0.75 \, m/s$
(i.e., a sink), is the maximum climb rate that can be achieved with no
thermal updraft. This climb rate is obtained in a straight descending
flight with $\sigma = 0°$ and $\alpha = 6°$. Naturally, for gliders, $v_z^{baseline} < 0$, hence, for
example, any positive $v_z^{agent}$ indicates positive soaring efficiency.
Overall, $\eta$ represents the fraction of the updraft exploited by the agent.
An agent with $\eta = 1$ uses all the thermal updraft; an agent with $\eta = 0$ can
be gliding straight at $v_z^{baseline}$; and an agent with $\eta < 0$ is doing worse
than $v_z^{baseline}$.

Figure 3b shows the distributions of the agent's mean $v_z$ value for
different horizontal winds, compared with $v_z^{baseline}$ and $v_z^{optimal}$, and
Fig. 3c plots the corresponding $\eta$ distributions. The agent achieved
$\eta > 0$ across the entire range of horizontal wind speeds and performed
best in the mid-range $u = 1 - 3 \, m/s$, with $\eta \approx 0.8$. The highest average $v_z$
was $0.42 \, m/s$ obtained at $u = 3 \, m/s$. For this $u$, when omitting the first
$20 \, s$ of each trajectory, the mean $v_z$ reaches $0.54 \, m/s$ (maximum of
$0.67 \, m/s$), the mean $\eta$ reaches $0.88$ (maximum of $0.96$).

## Identifying learning bottlenecks via reward shaping
To reveal the underlying structure of the thermal soaring problem and
break it into meaningful sub-problems, we tested the nominal agent in
a slightly more challenging protocol, in which $u$ was randomly selected
between 2.5 and 3.5 m/s, without curriculum learning. We identified
that in our system the task of efficient thermal soaring is too difficult
for learning when the reward is simply the climb rate $v_z$. As shown in
Fig. 4a, an agent with this reward did not manage to stabilize itself and
crashed in a very early stage of the simulation. The first sub-problem of
thermal soaring is, therefore, achieving stable flight. Hence, we chan-
ged the reward function by adding the penalty $P_{stab}$, defined above
(Thermal soaring under horizontal wind in Results section), which
penalizes the agent when it gets unstable. Agents that learned with the
reward $v_z + P_{stab}$ achieved stable flight (Fig. 4a) but did not manage to
find the thermal and to spend a significant time in its vicinity (Fig. 4b).
This reveals the second sub-problem of thermal soaring – staying close
to the thermal center. Thus, we defined another penalty term, $P_{center}$,
proportional to the agent's distance from the thermal center (Thermal
soaring under horizontal wind in Results section). Agents learning with
the combined reward $v_z + P_{stab} + P_{center}$ managed to fly both stably and
near the thermal center, and eventually learned to circle the center to
soar efficiently (Fig. 4c). Interestingly, agents with the reward
$v_z + P_{center}$ did not overcome the stabilization bottleneck (Fig. 4a).

In summary, these results indicate that the problem of thermal
soaring, as modeled here, consists of at least two sub-problems that
can be manifested as learning bottlenecks (Fig. 4d). The first bottle-
neck identified is achieving stable flight, and the second is locating and
flying near the thermal center. Using reward shaping assisted the agent
to sequentially funnel its flight technique through those bottlenecks
and achieve efficient soaring. Interestingly, these gliding principles,
which are known by human glider pilots, were "rediscovered" by the
current RL system using an NN with only 400 inter-neurons.

## Probing the state and action representations
As part of the state optimization process, we tested agents with several
state representations, whose performance shed light on the impor-
tance of specific attributes of the nominal state representation: the
horizontal wind speed $u$ and the memory size of previous states.

**Horizontal wind speed.** First, we compared the nominal agent to an
agent whose state representation excluded $u$, including only $\{V, v_z, \sigma, \alpha, \theta\}$. Both agents' memory included their current state and the state in
the previous 8 time steps, and they were trained in the nominal cur-
riculum learning protocol (Thermal soaring under horizontal wind in
Results section). Each agent was evaluated on a range of horizontal
winds $u = 0-6$ for 100 runs on each value of $u$. Figure 5a shows that the
agent whose state did not include $u$ was significantly less efficient than
the nominal agent which had $u$ in its state. Additionally, we found that
agents without knowledge of $u$ were able to learn to soar under one
specific horizontal wind $u_0 \leq 8 \, m/s$, when trained in a curriculum
learning protocol which gradually increased in 1 m/s steps. However,

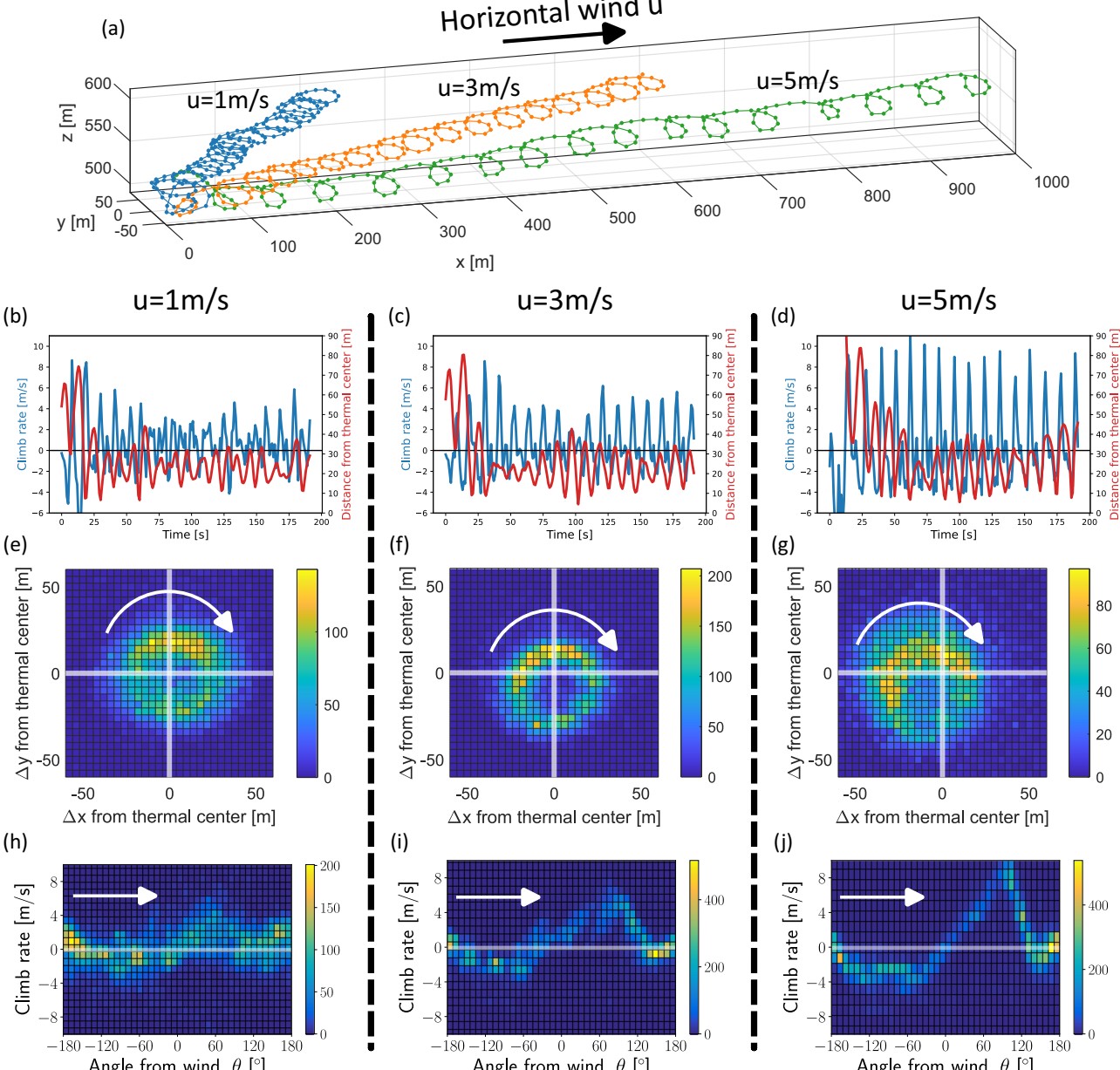

**Fig. 2 | Thermal soaring of the nominal agent under various horizontal wind speeds. a** Representative 3D soaring trajectories in winds of 1 m/s (blue), 3 m/s (orange) and 5 m/s (green). The wind is in the +x direction. **b–d** The climb rate $v_z$ and the agent's distance from the (moving) thermal center as a function of time, for the same trajectories in (**a**). The left column is for $u = 1$ m/s, middle column is for $u = 3$ m/s, and the right column is for $u = 5$ m/s. **e–g** Histograms of the agent's $xy$ position with respect to the the moving thermal center. Each histogram for a given value $u$ was calculated for 100 runs with random initial conditions. The white arrows indicate the direction of circling. **h–j** 2D histograms of the climb rate $v_z$ and the angle from wind, $\theta$. The histograms were calculated for the same 100 runs per each $u$. The white arrows indicate the direction of circling.

these agents performed poorly on horizontal winds weaker than $u_0$, indicating that this configuration was over-fitted for a specific horizontal wind. Together these results demonstrate that $u$ is a crucial state parameters, and hint that an agent without explicit knowledge of $u$ would have to infer it from other information (see the Discussion section).

**Memory size.** Next, we compared the nominal agent to a series of agents whose state representations included memory buffers of different sizes, from 1 to 9 previous time steps (nominal is 8), in addition to the current state. These agents were trained under horizontal winds of 1–3 m/s and evaluated on $u = 2$ m/s. Figure 5b shows that for staying

mostly in the vicinity of the thermal it is sufficient to have memory of ≥2 time steps. However, for exploiting the thermal and soaring efficiently it is required to have memory of ≥5 time steps, as measured by $\eta$. Interestingly, inferring the center position of a thermal updraft from local updraft observations is a non-Markovian problem. A previous observability analysis of thermal soaring[35], has also shown that a history of ≥2 time steps is sufficient for this inference.

**Actions.** Our model allows for three control parameters: $\Delta\sigma$, $\Delta\alpha$, and $\Delta\beta$, which represent modulations the the bank angle, angle-of-attack and sideslip angle, respectively. While the bank angle is crucial for thermalling[4,8,9,41], it is insufficient for stabilizing in strong horizontal

(a)

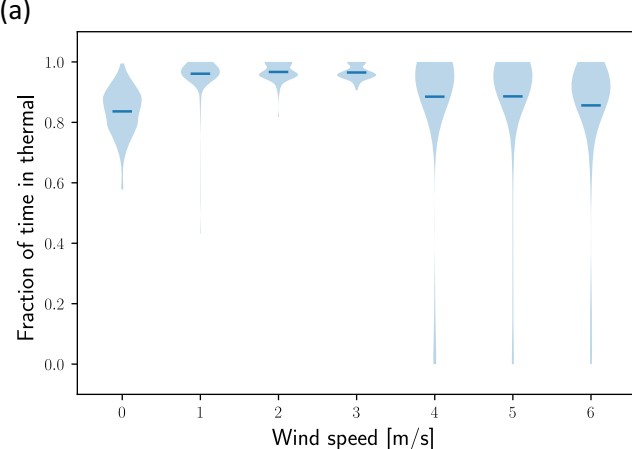

(b)

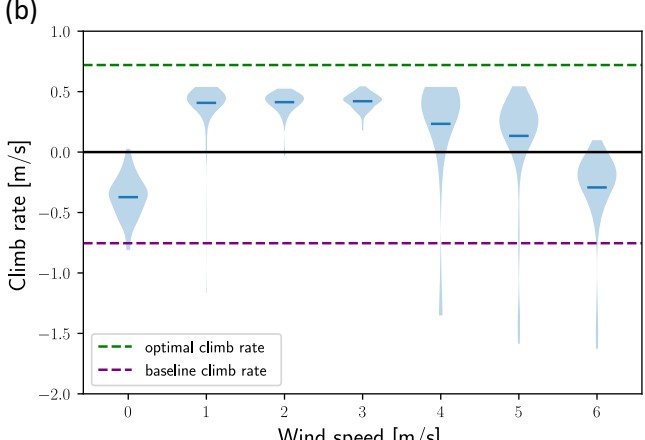

(c)

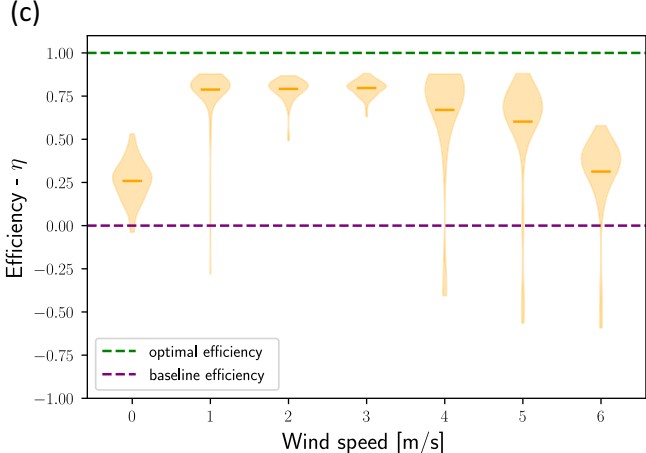

**Fig. 3 | Thermal soaring performance of the nominal agent at different horizontal wind speeds.** Agent's performance was analyzed over 100 runs with random initial conditions for each value of the horizontal wind speed $u$. The performance measures are presented in violin plot which show the distribution overall runs with the mean marked by a horizontal line. For reference, each plot shows also the optimal and baseline climb rates (Supplementary Note 2). **a** The time the agent spent within the thermal as a fraction of its full run of 200 s. **b** The distribution of average climb rates for each $u$. **c** The distribution of the thermal soaring efficiency $\eta$ for each $u$ (Eq. (1)).

winds. As taught in basic gliding courses[4], our system consistently showed that adding $\Delta\alpha$ was sufficient for achieving efficient thermalling, whereas further adding $\Delta\beta$ resulted in slow convergence. Therefore, the actions of the nominal agent we set to $\Delta\sigma$, $\Delta\alpha$.

### Robustness of the nominal agent
To mimic several aspects of the real world, we characterized the robustness of the nominal solution with respect to changes in the parameters of the thermal and to sensor noise.

**Parameters of the thermal.** The nominal agent was trained on environments with random horizontal wind and initial condition but with fixed thermal parameters, $w^*$ and $z^*$ (Simulation Model in Methods section). To evaluate the agent's ability to generalize its technique for different thermal parameters, we tested its performance in a range of $w^*$ and $z^*$ values without further training (Fig. 6a). For each combination of $w^*$ and $z^*$ we ran 20 simulations with horizontal wind of $u = 3$ m/s and random initial conditions. The agent performance was evaluated by the efficiency metric $\eta$ and by calculating its average $v_z$ and optimal $v_z$ under each condition. Figure 6a shows that for the weaker thermals with $w^* = 3$ m/s the agent performed quite poorly, but it managed to generalize for all the tested values of $w^* \geq 5$ m/s and all $z^*$, where its efficiency was >70%.

**Sensor noise.** We modeled sensor noise by adding Gaussian noise to the state variables $\{v_z, V, \sigma, \alpha, \theta, u\}$, one variable at a time. Figure 6b shows the average climb rate of the nominal agent under different noise levels. The noise levels were normalized with respect to the standard deviation of the sensor readout, as a typical scale for its variation. Overall, the climb rate reduced with the noise level, with different performance degradation for noise in different sensors. For example, the most sensitive sensor was $\theta$ (angle with respect to $u$): relative noise of ~0.65 degraded the agent's performance down to the baseline climb rate, equivalent to gliding with no thermal. The least sensitive sensor was $V$, for which a similar performance degradation was met for relative noise of ~1.7.

Testing the performance under sensor noise in $u$ (Fig. 6c), shows that the agent's performance degraded to the baseline climb rate at a Gaussian noise with standard deviation of ~7 m/s, greater than the horizontal wind speed $u = 3$ m/s used in this trial. This result hints that the agent averages the readouts of $u$ across its memory buffer. Finally, we tested the agent's robustness to environmental noise in $u$, which simulated a horizontal wind with gusts that change every 20 s and have a Gaussian distribution. The agent performance monotonically degraded to the baseline at gusts with standard deviation of ~3 m/s, where the mean $u$ was 3 m/s.

### Opening the black box: analyzing the neural network's functional modes
The highly complex and opaque nature of NNs makes it difficult for us to understand the resulting gliding policy. To address this challenge, we analyzed the NN activity by clustering the neural activation values in the hidden layers, to obtain clusters with similar activation patterns (Interpreting and analyzing the deep-RL agents in Methods section). We analyzed a nominal agent that was trained in a range of horizontal winds 1–2 m/s, whose performance during the learning process is shown in Fig. 7a. We selected three copies of the agent during the learning process and labeled them "young", "intermediate" and "expert". Representative trajectories of these three stages are show in Fig. 7b. Clustering the NN activation patterns revealed four distinct clusters. At any given time within a flight trajectory we associated the internal NN activation pattern with its closest cluster and obtained a labeling of each timepoint according to its cluster (color coded in Fig. 7b).

Markedly, this cluster labeling can be associated with specific stages of the circling flight in the thermal. Labeling is mostly associated with the angle of the glider with respect to the horizontal wind, which is equivalent to its phase in the circling motion (Fig. 1f). Furthermore, this association becomes more apparent with increasing agent's "age": Fig. 7c, which plots how cluster labeling is distributed by $\theta$, shows that

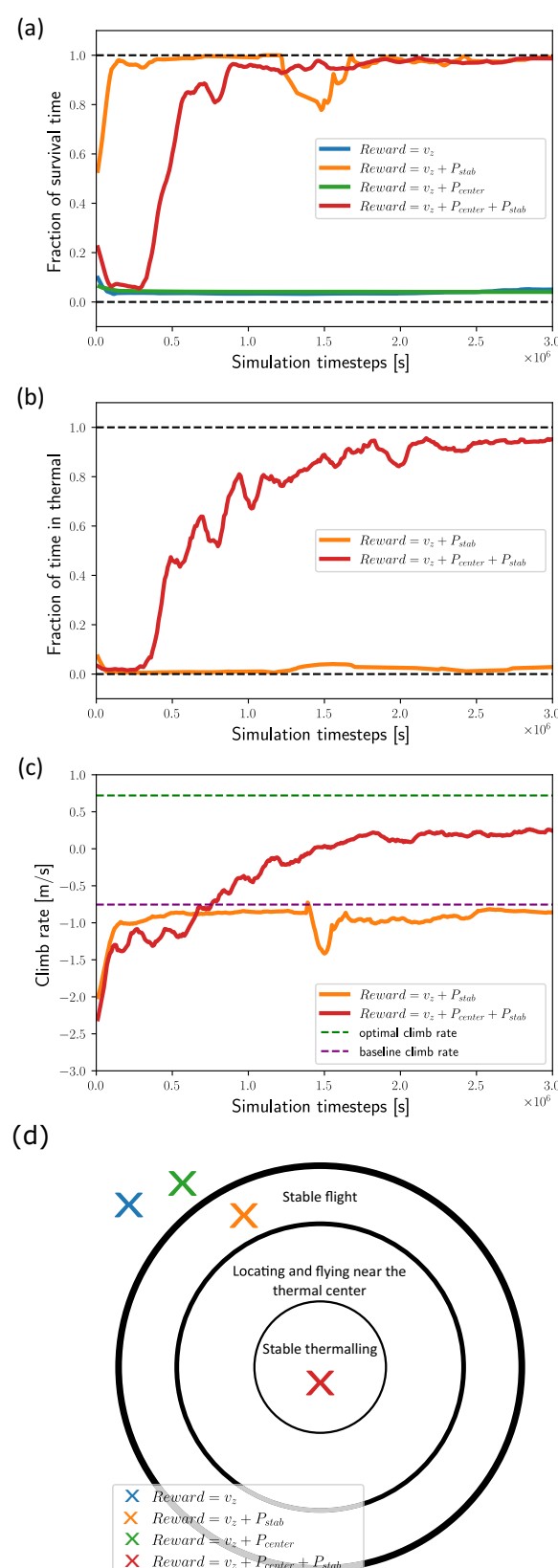

(a)

(b)

(c)

(d)

**Fig. 4 | Learning bottleneck.** Gliding performance during the learning process as a function of simulation time in seconds. The four colors represent agents with different reward functions (legend). **a** The fraction of stable flight time vs. learning time for different reward agents. **b** The fraction of time spent in the thermal area vs. learning times for different rewards for which agents that survived a significant time. **c** Climb rate vs. learning times for different reward functions, in which agents that survived for a significant duration. **d** Learning bottlenecks are represented as concentric circles of flight technique that should be learned to achieve stable thermalling (inner circle). The × symbols indicate the stages reached by the agents in (**a**–**c**) with the same color coding. The transition into the outermost circle represents the first learning bottleneck related to stabilization. The transition to the next inner circle corresponds to the second bottleneck: locating and flying near the thermal. The final transition into the innermost circle of "stable thermalling" was achieved via a smooth process facilitated by the $v_z$ reward.

−100°. A fourth cluster (green) appeared for a brief time in the beginning of each trajectory and is associated with a short thermal searching phase. The changes in the clustering patterns over the agent's "lifetime" are reminiscent of the differences between the thermal soaring of young and adult vultures, where the climb rate of young vultures is independent of $\theta$ and the (significantly higher) climb rate of adult vultures depends on $\theta$[1]. To rule out the possibility that this functional association of the clusters with $\theta$ is a trivial outcome of having $\theta$ as a state variable, we performed the same analysis on an agent that did not have access to $\theta$ in its state (Supplementary Note 4). While this agent did not perform as well as the nominal agent, it did exhibit neural activation clustering with similar distribution in $\theta$.

**Comparing the RL agent to soaring vultures data**

Finally, to compare the performance of our RL agent to the motion of free-ranging soaring vultures that was measured by ref. 1, we selected 243 characteristic thermalling trajectories with consistent motion of ≥100 sec each, encompassing a total of 9.4 h, that were taken during a narrow time period (August–September). The magnitude and direction of the horizontal wind, as well as the bank angle, were estimated from the data. To adjust the parameters of the thermals ($z^*$, $w^*$) to be used during learning, we applied the calculation of $v_z^{optimal}$ (Comparing the RL agent to soaring vultures data in Methods section and Supplementary Note 2), while assuming that vulture climb in an optimal $v_z$. Then, we chose the values of ($z^*$, $w^*$) that correspond to the vultures' mean $v_z$ and thermalling radius. Using these parameters, we trained a nominal agent with vulture-like mass $m = 7.75$ kg and wing area $S = 0.87$ m² and tested it over 100 random trials.

Comparison shows conspicuous similarity between the vulture and agent thermalling trajectories. The distributions of circling radii (Fig. 8a, b) of the vultures and agent have similar average of ~30 m, where the agent's distribution is narrower, probably owing to the synthetic simulation conditions. The vultures' distribution of $\theta$ is more structured than for the agent, but both peak around 180°, when flying along the horizontal wind (Fig. 8c, d). Figure 9 shows that the mean values of the speed $V$, climb rate $v_z$ and bank angle $\sigma$ of both vultures and agent are comparable.

## Discussion

We developed a simulation-based deep-RL system that learned to perform efficient thermal soaring under strong horizontal wind, and we used this system to address questions about the learning of motion control. Modulating the reward function revealed an underlying structure of the thermal soaring problem in its current formulation. This problem consists of at least two bottlenecks, or sub-problems, that must be solved sequentially to achieve efficient soaring: the first bottleneck is achieving stable flight and the second is flying at the vicinity of the thermal center. While using a reward that penalizes the agent based on its distance from the thermal center is applicable only in simulations and not during training in the real world, such a penalty

as the agent "matures", the distribution of clusters becomes more correlated with $\theta$. For example, in the "expert" agent, one cluster (red) is correlated with flight against the wind in the range $\theta = -100°$ to 90°, a second cluster (purple) is correlated with the values from $\theta = 90°$ to ±180°, and a third cluster (blue), aligns with the range $\theta = ±180°$ to

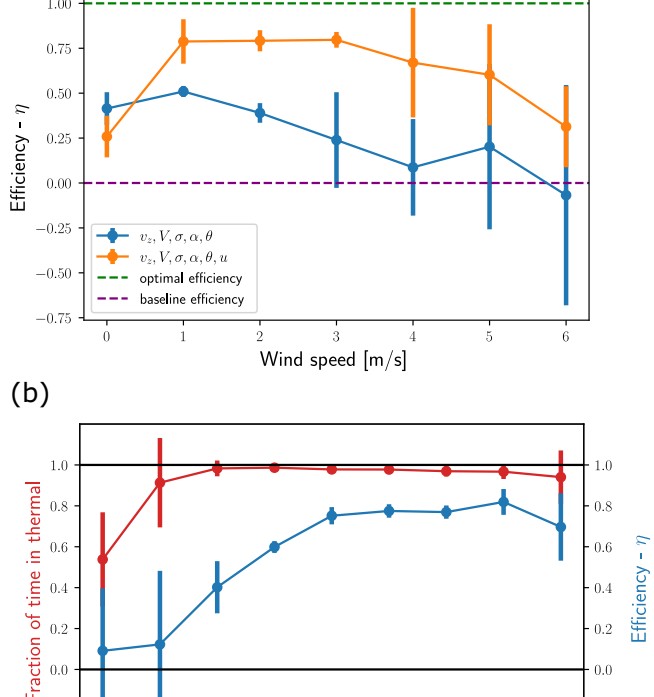

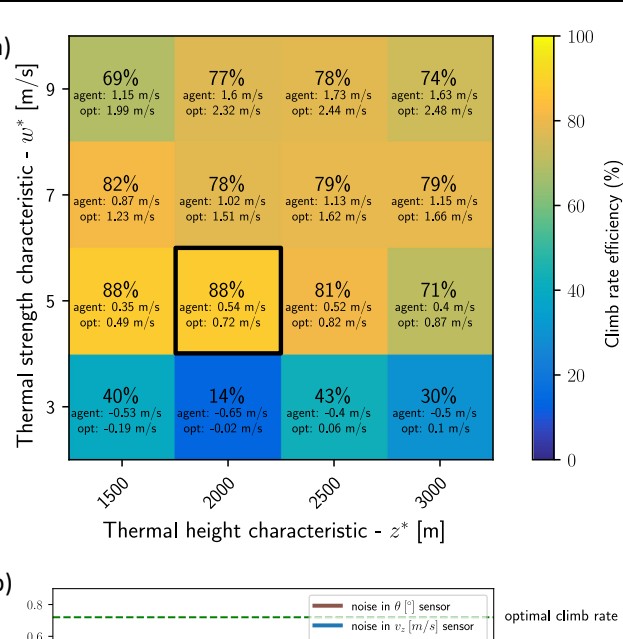

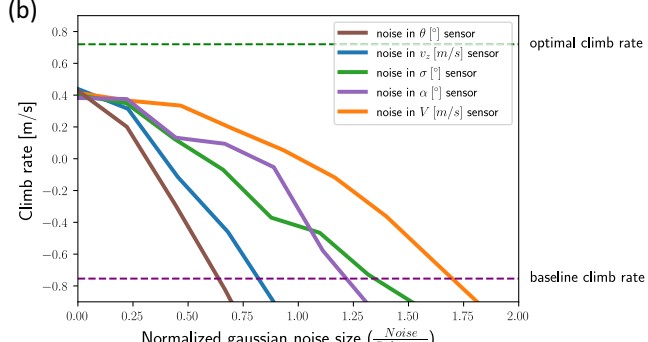

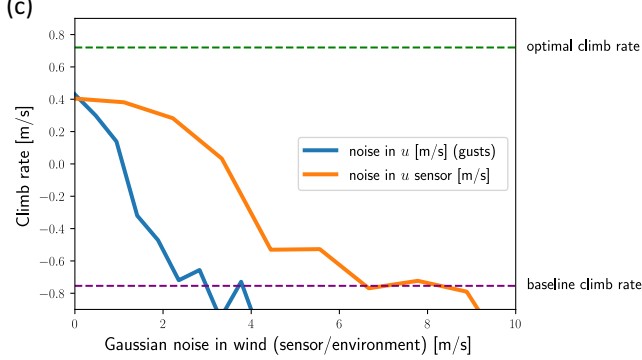

**Fig. 5 | State representation. a** The horizontal wind $u$ is a crucial state variable. The thermalling efficiency $\eta$ was calculated for two agents under the same learning protocol. One agent (orange) had $u$ included in its state, and the other agent (blue) did not. The thermalling efficiency is plots as the mean $\eta$ on 100 runs in each horizontal wind speed $u$ from 0 to 6 m/s. For reference, the baseline and optimal values of $\eta$ are plotted as well. **b** Agent performance as a function of its memory buffer size. The efficiency (blue) and fraction of time spend within the thermal (red) are plotted for the agents trained under $u = 1 - 3$ m/s and tested on $u = 2$ m/s. These results show that a memory buffer of 3 s is sufficient for flying within the thermal, but efficient soaring requires a buffer of at least 5 s.

may be of some value. Pre-training real-world agents in simulation with such a penalty may improve their ability to estimate the thermal position based on their available state information, thereby bridging some of the "sim-to-real" gap.

Identifying the learning bottlenecks and mitigating them using reward shaping enabled us to train a small NN to soar in relatively strong winds. The method of reward shaping may be applicable for finding the underlying structure of other complex problems in RL. Identifying learning bottlenecks and addressing them sequentially during learning can, therefore, improve the solution strategy and convergence time for problems that are too difficult for a direct solution based on a single final goal. This is similar to learning to crawl before learning to walk and run. Additionally, we showed that curriculum learning was crucial for learning to soar under different horizontal wind speeds. The principle of gradually learning to handle more difficult conditions may be applicable for other motion control problems.

Using a new efficiency metric, $\eta$ (Eq. (1) and Figs. 3, 5a, 6a), which quantifies how well the thermal is exploited by the agent, we characterized the robustness of the acquired policy with respect to environmental conditions, sensor noise and different state–action representations. The control over the environment, state, action, and reward offered by such RL system, may be helpful in gaining insights into the limiting factors of complex behaviors such as thermal soaring.

**Fig. 6 | Robustness of the nominal agent. a** Climb rate performance in different thermal parameters. To quantify the system's generalization, the nominal agent was tested on 15 combinations of thermal parameters $(z^*, w^*)$ different than the values on which it was trained (dark square). For each combination of $(z^*, w^*)$ we averaged over 20 runs in random initial conditions. The plot shows the mean efficiency $\eta$ in percent (color coded), the mean $v_z$, and $v_z^{\text{optimal}}$. **b**, **c** Climb rate performance degrades with sensor noise and wind gusts. **b** The average climb rate of the nominal agent under different types of sensor noise and different normalized noise levels. The noise level in each sensor was normalized by the standard deviation of the measured variable. The standard deviations were: std($V$) = 2.3 m/s, std($v_z$) = 2.6 m/s, std($\theta$) = 116.4°, std($\sigma$) = 11.9°, std($\alpha$) = 5.9°. **c** The orange line describes the agent's average climb rate as a function of noise level in its $u$ sensor, as defined in the main text. The blue line shows the average climb rate as a function of noise in the environmental value of $u$, which represents physics gusts that change over a 20 s timescale.

This methodology is widely applied for non-NN, models[42] and has recently been applied for deep learning and RL models of the sensory system of bats[40,43]. Here, for example, by testing various action representations we found that controlling the glider's bank angle alone

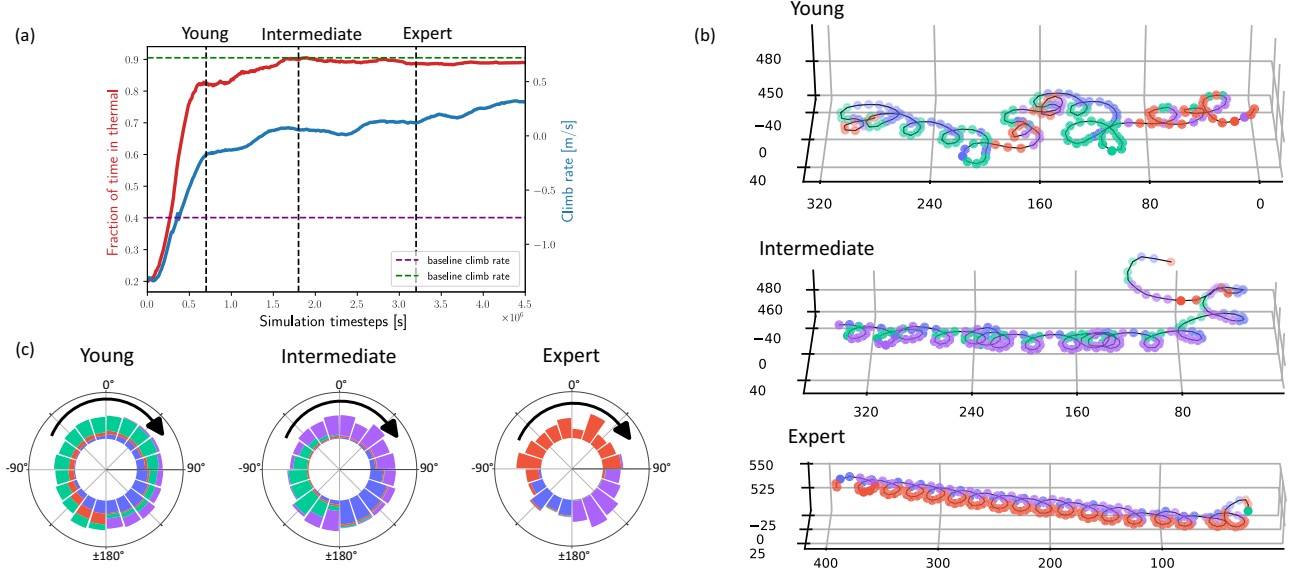

**Fig. 7 | NN activation clusters through learning process. a** The evolution of agent performance during the learning process. Dashed vertical lines mark the times at which the "young", "intermediate", and "expert" agents were selected for further analysis. **b** Representative trajectories of the three agents colored the active neural cluster at each timestep. **c** Distribution of the neural activation clusters as a function $\theta$ for each of the three agent. Color coding is identical to (**b**).

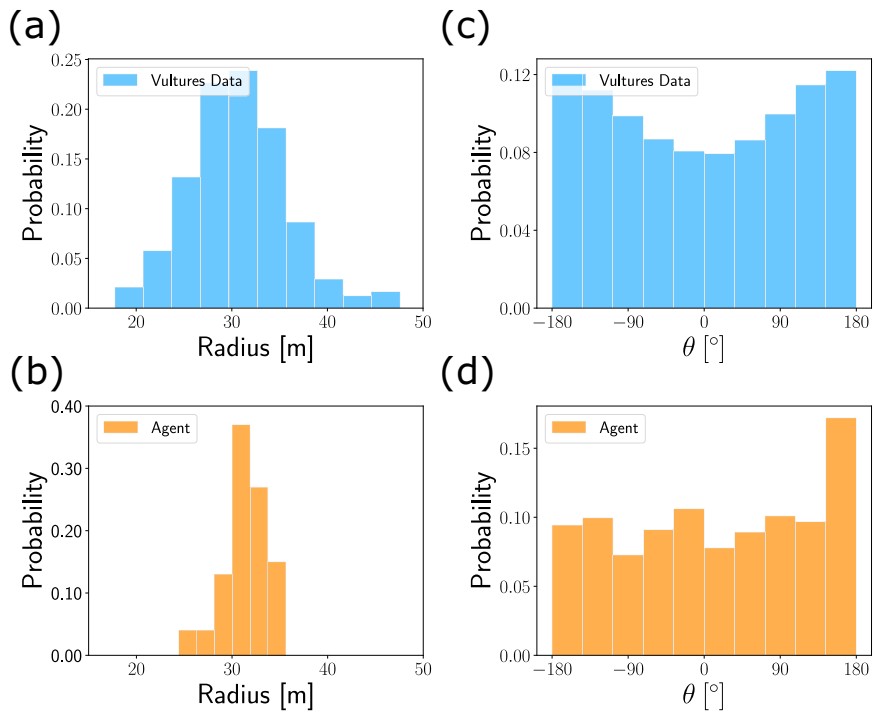

**Fig. 8 | Comparing the RL agent and thermalling vultures. a**, **b** Distributions of the thermalling radii for the vultures (**a**) and agent (**b**). **c**, **d** Distributions of, $\theta$, the angle with respect to the horizontal wind for the vultures (**c**) and agent (**d**).

is insufficient for stable flight under strong horizontal winds, and that the angle-of-attack must be controlled as well. Testing different state representations highlighted the importance of the wind speed $u$ and angle $\theta$ for efficient soaring. These results raise the questions of whether and how these parameters are measured or estimated by soaring birds, and how they can be estimated by UAVs. If not measured directly, can these parameters be observable by a given flying system, *e.g.*, by using a body-mounted airspeed sensor combined with a calculation that exploits body rotation[35,44]. characterizing the agent's robustness to sensor noise and wind gusts may provide physical limits to the relative importance of different sensory information in soaring birds. Further, comparing the flight trajectories of the RL agent performance to the trajectories of soaring vultures together with the efficiency metric, learning bottlenecks and the evolution of the agents technique with its "age", may shed light on the constraints and learning process of soaring birds. These concepts may be tested by long-term motion tracking of individual birds combined with accurate measurement of the local environment.

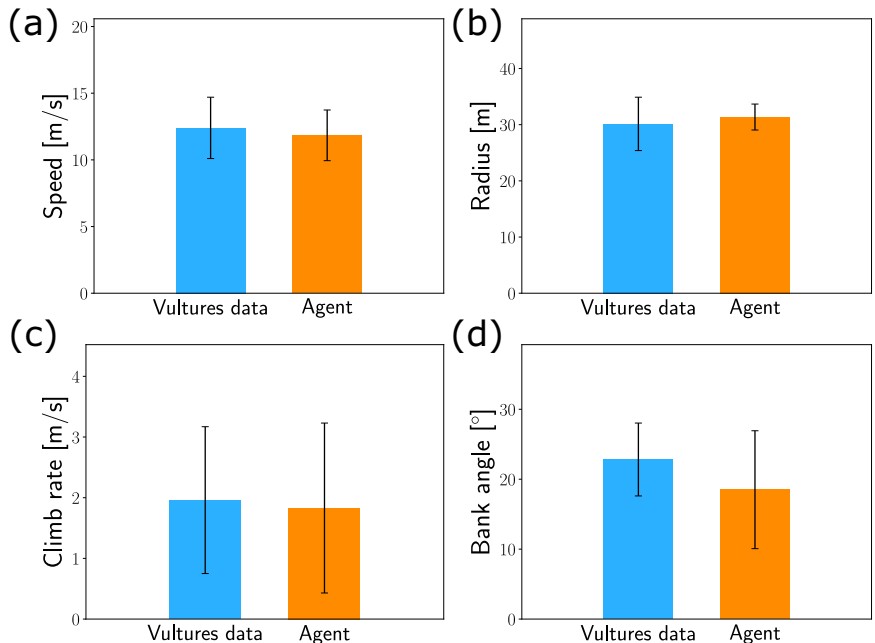

**Fig. 9 | Comparing the mean trajectory parameters of the RL agent and thermalling vultures.** Each bar represents the mean value of a kinematic parameter, with the whiskers indicating its standard deviation. **a** Speed $V$. **b** Thermalling radius. **c** Climb rate $v_z$. **d** Bank angle $\sigma$.

Finally, we analyzed the neural activation patterns of the agent's NN and found it divides into functional clusters. During the learning process, this clustering became more distinct, and in the "expert" agent the clusters were associated with specific phases of $\theta$ during the circling motion around the thermal center. Further analyzing these clusters and their evolution during the learning process may inform us about the inner workings of the NN and may be used to understand the agent's policy by extracting explicit rules, for example, by describing each cluster as a decision tree[45–48]. In summary, the application of deep-RL method as presented here may contribute to further improving autonomous UAV gliding systems and to our understanding of motion control learning.

## Methods

### Thermal soaring as an RL problem

Reinforcement learning is a type of ML in which an agent learns to take actions in an environment to maximize a cumulative reward signal (Figs. 1 and 10). The agent observes its state in the environment and chooses an action based on a policy $\pi$: $S \times A \to p$, where $S$ is the state space and $A$ is the action space. The policy $\pi$ maps state–action pairs to the probability $p \in [0, 1]$ of taking a particular action in a given state. The policy can be represented in various ways and here we represent it as a NN. The agent receives a reward based on the state resulting from the selected action. The goal of RL is to learn a policy that maximizes the expected cumulative reward. In our thermal soaring environment, the agent is a simulated glider with vulture-like parameters. The agent's kinematic parameters, such as speed and direction, represent its state, while the glider's control parameters, which are modulations of its aerodynamics angles, represent the actions. The reward signal can be designed to incentivize the agent to gain altitude, which in our simulated environment can be achieved by thermal soaring. Through trial and error, the goal of the RL algorithm is to shape the policy $\pi$ to optimize the agent's behavior in the thermal soaring environment based on the reward received.

### Simulation model

In our RL framework, the environment is a simplified three degrees-of-freedom simulation model of a small glider in the presence of wind,

used as an approximation for flight dynamics under moderate wind speeds. It is based on the simplified model of ref. 7, in which the glider is described as a point-mass aircraft in an environment with a spatially varying wind. The glider's parameters, such as mass, wingspan, and aerodynamic coefficients are detailed in Supplementary Note 1. The glider model is described in Fig. 1 and its equations of motion are:

$$
\begin{aligned}
\dot{z} &= V \sin \gamma \\
\dot{x} &= V \cos \chi \cos \gamma \\
\dot{y} &= V \sin \chi \cos \gamma \\
\dot{V} &= -\frac{1}{m} D(V, \alpha, \beta) - g \sin \gamma \\
\dot{\gamma} &= \frac{1}{mV} (L(V, \alpha) \cos \sigma + C(V, \beta) \sin \sigma) - \frac{g}{V} \cos \gamma \\
\dot{\chi} &= \frac{1}{mV \cos \gamma} (L(V, \alpha) \sin \sigma - C(V, \beta) \cos \sigma).
\end{aligned}
\tag{2}
$$

The dynamic variables are the $(x, y, z)$ coordinates of the glider's center-of-mass, its velocity $V$, glide angle $\gamma$, and side angle $\chi$. The glider's two control parameters are its bank angle $\sigma$ and angle-of-attack $\alpha$. The model supports a third control parameter – the sideslip angle $\beta$ – which, as we verified, is not essential for obtaining robust thermalling. The glider mass is $m$ and gravity acceleration is $g$. The lift force $L$, drag force $D$, and side force $C$, are calculated by a simplified aerodynamic model based on $V$, $\alpha$, and $\beta$, which include the effect of the local air motion due to thermals and horizontal wind. Detailed description of the aerodynamic model is given in Supplementary Note 1.

The environment's atmospheric model consists of a sum of a horizontal wind and thermal updraft. Horizontal wind is approximated as uniform wind velocity $u$ in the $+x$ direction. This direction is practically randomized with respect to the glider-agent, given that the agent has no prior information about its heading and its initial heading is randomized. Soaring birds and human glider pilots obviously use more information about the environment[49]. The magnitude of the horizontal wind is constant per each simulation run. We assume that $u$ has no $z$ dependence despite its known logarithmic profile $u(z)$ in open landscapes, because gliders and vultures mostly soar at relatively high elevations where the boundary-layer resistance (that shapes the

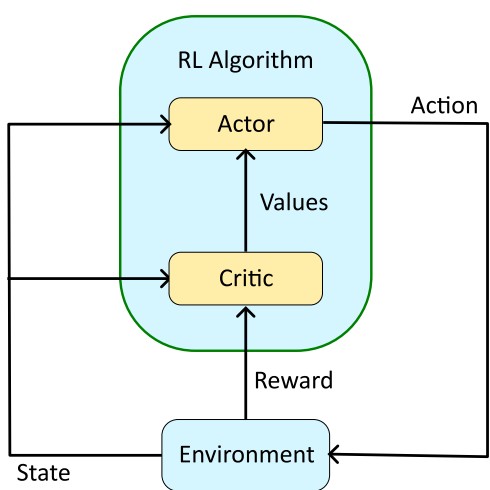

**Fig. 10 | Actor–critic RL architecture.** The actor acts in the environment and the critic learns the environment, by assigning a "value" to each state (or state–action pair). Critic values are used to update the actor. Together, the RL algorithm operates to obtain maximum cumulative reward.

logarithmic profile) is weak and $u(z)$ is nearly uniform. The shape of the updraft velocity vertical profile $w(z)$ is determined by Gedeon's model[50], and its amplitude $w_{core}(z)$ is given by Lenschow's model[51]. The combined vertical updraft velocity, as described by Bencatel is given in cylindrical coordinates $(r, z)$[52], is:

$$w(r,z) = w_{core}(z)\, e^{-\left(\frac{r}{R(z)}\right)^2}\left[1 - \left(\frac{r}{R(z)}\right)^2\right]$$

$$w_{core}(z) = w^*\left(\frac{z}{z^*}\right)^{1/3}\left(1 - 1.1\frac{z}{z^*}\right) \qquad (3)$$

$$R(z) = 0.08\left(\frac{z}{z^*}\right)^{1/3}\left(1 - \frac{1}{4}\frac{z}{z^*}\right)z^*,$$

where $w^* = 5\,\text{m/s}$ is the characteristic updraft amplitude and $z^* = 2000$ m is the characteristic thermal altitude. In the resulting radial updraft profile (Fig. 1c, d), the thermal radius increases with height $z$ and its velocity peak is surrounded by a ring of downwards velocity. For example, the peak updraft velocity at $z = 500$ m is $w_{core} = 2.3\,\text{m/s}$. To combine the horizontal wind and updraft velocity we use the chimney model[52,53], in which thermal motion may include both horizontal motion and leaning of its centerline. The results presented here use uniform horizontal motion without leaning. We achieved successful RL also with the full chimney model. The overall atmospheric model is, therefore:

$$\mathbf{v}(x,y,z,t) = \begin{pmatrix} u \\ 0 \\ w(r(t),z) \end{pmatrix}, \qquad (4)$$

with $r(t) = \sqrt{(x - ut)^2 + y^2}$ (Fig. 1d).

## Deep-RL model

The deep-RL agent consists of an NN that controls the simulated glider in an environment with a single thermal. Each simulation run spans 200 s of flight, divided into 1 s intervals. In each interval, the agent obtains its state and calculates a steering action. The action is implemented in the simulated dynamics of the next 1 s interval. In a real glider system, this mode of operation would rely on a nested feedback loop that provides orientation control via the aircraft control surfaces. After a full simulation run, the agent gets a reward based on its cumulative performance. The state, action and reward are described below.

**State.** The agent's state consists of the following parameters:

- Glider's speed $V$.
- Climb rate $v_z$.
- Bank angle $\sigma$.
- Angle-of-attack $\alpha$.
- The magnitude of the horizontal wind $u$.
- The angle $\theta$ between the glider's velocity vector and $u$ (Fig. 1f).

Each state entry was normalized to the range $[-1, 1]$. In gliders, these parameters can be measured by standard sensors, and can probably be sensed by vultures and other soaring birds. Additionally, the agent's state contains a short history of the state parameters, as vultures were found to integrate short-term information on their soaring performance from their current and recent experiences[54]. The length of this history is a hyper-parameter of the model. To characterize the robustness of our agents to noise, we tested some of the trained agents with sensor noise, where we added Gaussian noise to their state data (Robustness of the nominal agent in Results section). To further examine the effect of the state on performance, we tested additional state representations, which included the glider's distance from the thermal center, and the difference of the vertical wind between the two wings.

**Actions.** The agent's two actions are[7,9]:

- Bank angle change, $\Delta\sigma$. In each 1 s interval, the agent can change $\sigma$ by $\Delta\sigma \in [-15°, 15°]$, and $\sigma$ is limited to the range $[-50°, 50°]$.
- Angle-of-attack change, $\Delta\alpha$. In each timestep the agent can change $\alpha$ by $\Delta\alpha \in [-10°, 10°]$, and $\alpha$ is limited to $[-30°, 30°]$.

We also tested agents that can modulate their sideslip angle and wingspan, and found that these control parameters are not crucial for the current task.

**Reward.** In analogy to vultures and gliders, the basic reward in our model was the glider's climb rate $v_z$. To overcome the bottlenecks that we identified in the learning process (Identifying learning bottlenecks via reward shaping in Results section), we employed two additional factors in the reward, which can be considered as "reward shaping"[55]. The first factor is a penalty for instability, $P_{stab}$: when the agent spins out of control, the simulation is stopped and the agent is penalized proportionally to the remaining simulation time. The second reward shaping factor is a penalty proportional to the glider's horizontal distance from the thermal center, termed $P_{center}$. Importantly, because this penalty is not part of the state, the agent has no direct information on its distance from the thermal center. This penalty helps in shaping the agent's behavior through the reward, by encouraging the agent to fly closer to the thermal center (Identifying learning bottlenecks via reward shaping in Results section).

**Training algorithm.** We used the DDPG algorithm, which is a state-of-the-art actor–critic policy gradient method designed for continuous control problems[31] (Fig. 10). Actor–critic systems are composed of two NN: the actor network and the critic network. The actor determines the agent's action based on the current state, while the critic estimates the value or Q-functions associated with a given state and action. Both the value function and Q-function represent the expected cumulative future reward: the value function's estimate relies on a given state, while the Q-function's estimate relies on a given state and action. The critic network learns to estimate one of these functions by observing trajectories of (state, action, reward) generated by the actor in the environment. Most of the time, the critic's estimate is utilized to calculate the temporal difference (TD) error or advantage function. The TD error quantifies the discrepancy between the current estimate of the state-value and the discounted value estimate of the subsequent state, along with the actual reward obtained by the actor. The

advantage function is the expectation of the TD error with respect to the next state, which also quantifies the quality of a certain action given a certain state. The critic's value function serves the dual purpose of updating the critic's value estimation and providing feedback to the actor regarding the quality of its chosen action. In the case of DDPG, the critic network specifically calculates the Q-function rather than the value function. In addition to DDPG, we tested the proximal policy optimization (PPO) algorithm[30], which is another actor–critic policy gradient method. PPO also achieved successful learning, though in longer times.

We use the code implementation of the stable-baselines3 library[56], which includes various deep-RL algorithms based on the PyTorch library. We optimized the learning process by performing a hyper-parameter optimization[57] for the learning rate, learning algorithm (DDPG/PPO), number of NN layers, and layer size (Supplementary Note 3). Overall, in the learning process, we begin with an initial NN policy that maps states to actions. This policy is updated using a critic NN through the policy gradient actor–critic method, which is executed in the thermal environment and uses the rewards obtained from the environment.

In each simulation run during the learning process, the glider was initialized at a random orientation, random horizontal position in ($x$, $y$), height of $z = 500$ m, velocity of 15 m/s, and gliding angle of −5°. The thermal center was set to ($x$, $y$) = (0, 0), and the horizontal wind velocity $u$ in the $+x$ direction was set to a random value between $u_{min}$ and $u_{max}$, which are the wind speed hyper-parameters. The thermal was, therefore, drifting at velocity $u$ while keeping a vertical structure. These random initialization improved the agent's generalization capacity. On top of this scheme, we used a curriculum learning approach, where the maximum horizontal wind velocity, $u_{max}$, was gradually increased during the learning process.

### Interpreting and analyzing the deep-RL agents

To understand the agent's policy and performance during and after the learning process, we employed both kinematic analyses of the flight trajectories, and analyzed the NN itself by clustering the neural activation patterns in the network's hidden layers.

The activation of a neuron in the network is the output of its nonlinear activation function, which, in our case, is a non-negative number. At each timestep, we represent the activation of the entire network by concatenating the activations of all neurons in the NN hidden layers into one vector. Prior work has shown that analyzing the hidden layers is useful for interpreting NN in both physical-ecological[40] and RL problems[39]. To perform the clustering, we used $k$-means clustering[58] on the concatenated activation vectors. To determine the number of clusters $k$, we used the knee locator method[59] based on the sum-of-squared-error (SSE)[60], which finds the $k$ value at which the SSE curve starts to straighten up. We later show that specific clusters can be assigned to distinct flight modes.

### Comparing the RL agent to soaring vultures data

We compared our RL model to vulture flight trajectories collected by ref. 1. The dataset consists of over 4 million GPS location samples, taken at 1 Hz rate of 20 vultures of different ages collected over ~3 years. Among these locations, there were almost 1 million samples from thermal soaring in over 6000 thermal updrafts, with each thermal soaring trajectory lasting more than 100 sec. Selection of thermal trajectories is described in Comparing the RL agent to soaring vultures data in Results section.

To estimate the bank angle from the data, we consider the forces in the radial and vertical axes:

$$ma_r = L\sin(\sigma)\cos(\gamma)$$
$$mg = L\cos(\sigma)\cos(\gamma) - D\cos(\sigma)\sin(\gamma), \quad (5)$$

where $a_r$ is the radial acceleration, and assuming no vertical acceleration. Since $\sin(\gamma) = v_z/V \approx 0.1$ and $D < L$, we neglect the drag components in Eq. (5), and by dividing the remaining terms we estimate $\sigma \approx \arctan(a_r/g)$. We obtain $\sigma$ in each timestep by calculating $a_r$ from the data, and then averaging $\sigma$ over each trajectory. To validate the estimation the average $\sigma$ we tested this calculation on the simulation results, where $\sigma$ is known. On average, our estimation was accurate within 7%.

To approximate the wind velocity in the data, we calculated the average velocity in the $xy$ plane during an entire thermalling trajectory. This approximation neglects the effect of thermal leaning[52], yet it provides a wind velocity distribution similar to the distribution measured on the same dataset using the RAMS atmospheric model[1]. we use the estimated wind velocity to choose the data trajectories for comparison with the agent's trajectories. Estimation of the local thermalling radius is described in Estimating thermalling radius from trajectories in Methods section.

### Estimating thermalling radius from trajectories

We calculated the local thermalling radius of a soaring-flight trajectory of either the RL-agents or vultures. First, we projected the 3D trajectory points onto the 2D $xy$ plain. Then, we found the points where the trajectory crosses through $\theta = \pm 180°$ (flying with the wind, type A points), and $\theta = \pm 0$ (flying against the wind, type B points). We sorted these points according to their time-label and used them to form interlacing triangles of points ABA and BAB. Our estimate for the local thermalling diameter is the altitude in each triangle: in an ABA triangle we use the altitude from the B vertex, and in an BAB triangle we use the altitude from an A vertex.

## Data availability

Data generated in this study are available on https://github.com/MicroFlightLab/.

## Code availability

The code generated in this study is available on https://github.com/MicroFlightLab/.

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

## Acknowledgements

This study was supported by the Center for Interdisciplinary Data Science Research (CIDR) of the Hebrew University of Jerusalem, Israel (joint grant to T.B. and R.N.). R.H. was funded by the Deutsche Forschungsgemeinschaft (DFG, German Research Foundation) under Germany's Excellence Strategy, EXC 2117-422037984. The vulture study was funded by the US-Israel Binational Science Foundation (BSF 255/2008) and the special BSF Multiplier Grant Award from the Rosalinde and Arthur Gilbert Foundation (to R.N. and Wayne Getz). We also acknowledge financial support from the Adelina and Massimo Della Pergola Chair of Life Sciences and the Minerva Center for Movement Ecology (to R.N.). Finally, we thank Walter Nesser for his ecological insight and for taking us paragliding.

## Author contributions

Conceptualization and funding: T.B. and R.N. Methodology: Y.F., T.B. and A.T. Software, validation, analysis, visualization: Y.F. Writing: Y.F. and T.B. Data collection: R.H. and R.N. Supervision, resources and project administration: T.B.

## Competing interests

The authors declare no competing interests.
