## [Peer Review File · Nature Communications]

REVIEWER COMMENTS

Reviewer #2 (Remarks to the Author):

0. Summary

The authors present a learning-based approach to autonomous thermalling flight. The paper outlines the underlying reinforcement learning setup. The results showcase that the trained agent is capable of harvesting energy from thermal updrafts in a simulated environment under variable, challenging conditions. The principal contribution is the sound analysis of the trained agent's behavior alongside the comparison to the "learning dynamics observed in free-ranging vultures". From a technical perspective, the paper is very well written.

1. Major Comments

1.1. Claimed Contribution

The authors state that "the potential of deep-RL to the problem of thermal soaring under challenging wind is yet under fulfilled" and that the paper "present[s] a new simulation-based deep-RL solution to the problem of thermal soaring under challenging wind conditions." However, the recent article "Deep Reinforcement Learning Approach for Integrated Updraft Mapping and Exploitation" (<https://doi.org/10.2514/1.G007572>) addresses the same problem with a very similar approach. Moreover, that article presents actual flight test results of a soaring aircraft subject to a deep-learning-based policy under real-world "challenging horizontal winds." Interestingly, the real-world thermal flight trajectories presented in said article resemble the simulation results depicted in Fig. 6(a).

However, this does not render the contributions of the paper under review negligible. I suggest that the authors refer to the abovementioned article and focus on the (valid!) contribution of "analyzing the reward dynamics," "reveal[ing]" distinct neuronal patterns associated with specific behavioral models," and "identify[ing] similarities between the learned policy and the soaring technique and learning dynamics observed in free-ranging vultures."

1.2. Simulation Model

The authors state that their "deep-RL environment is a 6 degrees-of-freedom (DoF) simulation". In Sec. II.B., however, the paper presents a 3 (!) DoF model. While training in a 3DoF simulation environment is perfectly acceptable, the 3DoF model presented by the authors is simplified. The dynamics presented in Sec. II.B. (equation numbering missing) represent the standard 3DoF glider dynamics without wind. The authors state that "the aerodynamic forces are calculated in the wind frame of reference" (App. A). For once, it remains unclear what that means (lift, drag, and side force are aligned with the wind frame of reference by definition). Moreover, the model presented neglects that the air-relative velocity vector and

the flight-path velocity vector are not identical in the presence of wind. I think that the authors should refer to the model from Beeler, et al., "A Flight Dynamics Model for a Small Glider in Ambient Winds" for an unsimplified 3DoF model in the presence of wind and state that the model used in the paper under review is an approximation for moderate wind speeds.

2. Minor Comments

2.1. Control Variables

In Sec. II.A., the authors state that "the glider's control parameters, which are the body angles, represent the actions." I would not refer to bank angle and angle of attack as "body angles" but rather as "aerodynamic angles" (i.e., length and direction of the vector representing lift).

2.2. Non-Markovian Problem

The paper says that "the agent's state contains a short history of the state parameters" (Sec. II.C). I suggest that the authors state that inferring the center position of a thermal updraft from local updraft observations is a non-Markovian problem. The finding that "it is sufficient to have a memory of ≥ 2 time steps" (Sec. III.B.) is in line with the observability analysis presented in "Multiple Thermal Updraft Estimation and Observability Analysis" (<https://doi.org/10.2514/1.G004205>).

2.3. Figure 11

The units for $\text{std}(\theta)$ and $\text{std}(\omega)$ in the description of Fig. 11 appear wrong. Also, " $\text{std}(\omega)$ " shows up twice. Should read " $\text{std}(\theta) = [\dots]$, $\text{std}(\alpha) = [\dots]$ ", I suppose.

2.4. Reward Shaping

Personally, I am no big fan of reward shaping as it easily distorts the actual objective. Specifically, assigning the penalty term " P_{Center} " (Sec. III.D.) is only feasible for training on synthetic data as ground truth for the updraft center is hardly available in a real-world scenario. From my point of view, the authors should at least state that fact. Even more so, as the agent seems to be capable of learning to exploit an updraft when this penalty term is assigned (cf. Fig. 12)

3. Concluding Remarks

Despite the criticism stated above, I want to point out that, after minor revision, I recommend publication. The technical content is sound. Moreover, the insights into the learning dynamics and trained agent's behavior are interesting. Notably, the display of information in Fig. 6, 7, 10, and Fig. 14 is excellent.

Reviewer #3 (Remarks to the Author):

This manuscript presents a study of thermal soaring in a simulated environment with horizontal winds by means of deep reinforcement learning techniques. The paper is well written and appears to be technically correct.

However, the papers cited below have explored in depth this problem by a very similar approach, actually going beyond the content of this manuscript in at least two aspects

- 1) by using recurrent neural networks for memory based (and not just reactive) controls
- 2) testing in real world experiments

In conclusion, I think this is a good paper but does not meet the high editorial threshold for broad impact and significance required for publication in Nature Communications.

Notter, S., Schimpf, F., and Fichter, W., "Hierarchical Reinforcement Learning Approach Towards Autonomous Cross-Country Soaring," AIAA Scitech 2021 Forum, 2021. doi:10.2514/6.2021-2010.

Notter, S., Schimpf, F., Müller, G., and Fichter, W., "Hierarchical Reinforcement Learning Approach for Autonomous Cross-Country Soaring," Journal of Guidance, Control, and Dynamics, Vol. 46, No. 1, 2023, pp. 114–126. doi:10.2514/1.G006746.

Deep Reinforcement Learning Approach for Integrated Updraft Mapping and Exploitation

Stefan Notter, Christian Gall, Gregor Müller, Aamir Ahmad, and Walter Fichter

Journal of Guidance, Control, and Dynamics 2023 46:10, 1997-2004

Also note this recent paper with a closely related approach

Cui, Y.; Yan, D.; Wan, Z. Study on the Glider Soaring Strategy in Random Location Thermal Updraft via Reinforcement Learning. Aerospace 2023, 10, 834. <https://doi.org/10.3390/aerospace10100834>

Response to Referees

Manuscript: Revealing principles of autonomous thermal soaring in windy conditions using vulture-inspired deep reinforcement-learning

Overview: We are grateful to the reviewers for their insightful comments and suggestions. We addressed them all and changed the focus of the manuscript accordingly. We believe the comments have significantly improved the delivery of our results. A point-by-point response to the reviewers' comments is provided below. The revised manuscript is provided with edits highlighted (track-changes style) and line numbers, which we reference in this response. Since large parts of the text have been modified, a "clean" version, which incorporates all changes, is included as well. Overall, the manuscript was shortened and the introduction was rewritten, figures were grouped together by subject, and the technical parts were moved to a Methods section in the end of the text to improve readability and comply with the Nature Communications format. Additionally, we added new analysis comparing the RL-agent and the data measured from real vultures.

Comments from Reviewer 2:

Reviewer (comment 0): *The authors present a learning-based approach to autonomous thermalling flight. The paper outlines the underlying reinforcement learning setup. The results showcase that the trained agent is capable of harvesting energy from thermal updrafts in a simulated environment under variable, challenging conditions. The principal contribution is the sound analysis of the trained agent's behavior alongside the comparison to the "learning dynamics observed in free-ranging vultures". From a technical perspective, the paper is very well written.*

Response (comment 0): We appreciate the reviewer's opinion.

Reviewer (comment 1.1): *The authors state that "the potential of deep-RL to the problem of thermal soaring under challenging wind is yet under fulfilled" and that the paper "present[s] a new simulation-based deep-RL solution to the problem of thermal soaring under challenging wind conditions." However, the recent article "Deep Reinforcement Learning Approach for Integrated Updraft Mapping and Exploitation" (<https://doi.org/10.2514/1.G007572>) addresses the same problem with a very similar approach. Moreover, that article presents actual flight test results of a soaring aircraft subject to a deep-learning-based policy under real-world "challenging horizontal winds." Interestingly, the real-world thermal flight trajectories presented in said article resemble the simulation results depicted in Fig. 6(a).*

However, this does not render the contributions of the paper under review negligible. I suggest that the authors refer to the abovementioned article and focus on the (valid!) contribution of "analyzing the reward dynamics," "reveal[ing]" distinct neuronal patterns associated with specific behavioral models," and "indentify[ing] similarities between the learned policy and the soaring technique and learning dynamics observed in free-ranging vultures."

Response (comment 1.1): We thank the reviewer for highlighting the contributions of our work. As suggested by the reviewer, given the recent results presented in the 2023 papers by Notter *et al.*, and the 2023 paper by Cui *et al.* that was highlighted by Reviewer 3, we revised

the focus of the paper. Given the cumulative advancements in achieving thermal soaring using reinforcement learning (RL) system, both in simulations and in real gliders, the revised manuscript now focuses on leveraging these advancements to study thermal soaring as a model-system for motion control. For example, the revised introduction surveys the recent literature and presents several questions that we later address using our RL framework:

“The success of deep-RL models in handling thermal soaring opens the way for using such models to address basic questions related to the learning of motion control. These include, for example: what is the structure of the problem in terms of its bottlenecks – sub-problems that must be solved sequentially to achieve the final behavioral goal? How robust is the acquired policy? Can we dissect the agent's neural network – a computational object that is typically treated as a ‘black box’ – to gain insight into its function? And, how does a learned policy compare with the thermal soaring behavior of birds in the wild?” (Line 111 in the clean version, line 142 in the track-changes version).

The results section focuses on addressing these questions using our RL framework. Section A reports the performance of the RL system. Section B analyzes the learning bottlenecks that were identified in the learning process. Sections C and D probe the systems robustness to its inner representation and to environmental and sensor noise, Section E analyzes the inner working of the agent’s neural network and shows it consists of functional clusters, and Section F compares the performance of the RL agent to real vultures. The Abstract and Introduction were rewritten, the Results Section was edited and reordered, and the Discussion was significantly trimmed to focus on implications of the reported results for understanding the learning of motion control.

Reviewer (comment 1.2): *Simulation Model. The authors state that their "deep-RL environment is a 6 degrees-of-freedom (DoF) simulation". In Sec. II.B., however, the paper presents a 3 (!) DoF model. While training in a 3DoF simulation environment is perfectly acceptable, the 3DoF model presented by the authors is simplified. The dynamics presented in Sec. II.B. (equation numbering missing) represent the standard 3DoF glider dynamics without wind. The authors state that "the aerodynamic forces are calculated in the wind frame of reference" (App. A). For once, it remains unclear what that means (lift, drag, and side force are aligned with the wind frame of reference by definition). Moreover, the model presented neglects that the air-relative velocity vector and the flight-path velocity vector are not identical in the presence of wind. I think that the authors should refer to the model from Beeler, et al., "A Flight Dynamics Model for a Small Glider in Ambient Winds" for an unsimplified 3DoF model in the presence of wind and state that the model used in the paper under review is an approximation for moderate wind speeds.*

Response (comment 1.2): The description of the simulation model was changed as follows:

“In our RL framework, the environment is a simplified 3 degrees-of-freedom simulation model of a small glider in the presence of wind, used as an approximation for flight dynamics under moderate wind speeds.” (Line 604 in the clean version, line 779 in the track-changes version). And similarly, in the Appendix (lines 1045/1290, respectively).

The statement about calculating the aerodynamic forces in the wind frame-of-reference was removed to avoid confusion (Appendix, line 1299 in the track-changes version)

This is related to the reviewer's next comment that "*Moreover, the model presented neglects that the air-relative velocity vector and the flight-path velocity vector are not identical in the presence of wind.*". Indeed, the equations that appear in the manuscript describe, for simplicity, a model that does not include external wind, following the first and simplified model proposed by Beeler *et al.*. Importantly, in our simulation we did use the full model from Beeler *et al.*, which includes wind effects. Hence these effects are not neglected in our numerical framework. Delivering this point was the purpose of the above (confusing and then, deleted) sentence stated that the forces were calculated in the wind frame of reference. To clarify, we added the following sentence:

"Following Beeler *et al.*⁷, to account for the effects of wind on the glider, the full aerodynamic model calculates the forces in the wind frame-of-reference and then transforms these forces to the world frame-of-reference for solving the equations of motion." (Appendix, line 1054 in the clean version, line 1300 in the track-changes version)

Equation numbering was added to each one of the components of the force model.

Reviewer (minor comment 2.1): *Control Variables. In Sec. II.A., the authors state that "the glider's control parameters, which are the body angles, represent the actions." I would not refer to bank angle and angle of attack as "body angles" but rather as "aerodynamic angles" (i.e., length and direction of the vector representing lift)?*

Response (minor comment 2.1): The text was changed accordingly

"...the glider's control parameters, which are modulations of its aerodynamics angles, represent the actions." (Line 595 in the clean version, line 770 in the track-changes version).

Reviewer (minor comment 2.2): *Non-Markovian Problem. The paper says that "the agent's state contains a short history of the state parameters" (Sec. II.C). I suggest that the authors state that inferring the center position of a thermal updraft from local updraft observations is a non-Markovian problem. The finding that "it is sufficient to have a memory of ≥ 2 time steps" (Sec. III.B.) is in line with the observability analysis presented in "Multiple Thermal Updraft Estimation and Observability Analysis" (<https://doi.org/10.2514/1.G004205>).*

Response (minor comment 2.2): As suggested, we added the following:

"Interestingly, inferring the center position of a thermal updraft from local updraft observations is a non-Markovian problem. A previous observability analysis of thermal soaring²⁹, has also shown that a history of ≥ 2 time steps is sufficient for this inference." (Line 359 in the clean version, line 456 in the track-changes version).

Reviewer (minor comment 2.3): *Figure 11. The units for $\text{std}(\theta)$ and $\text{std}(\omega)$ in the description of Fig. 11 appear wrong. Also, " $\text{std}(\omega)$ " shows up twice. Should read " $\text{std}(\theta) = [\dots]$, $\text{std}(\alpha) = [\dots]$ ", I suppose.*

Response (minor comment 2.3): Corrected (previous Fig. 11 is current Fig. 6). With thanks.

Reviewer (minor comment 2.4): *Reward Shaping. Personally, I am no big fan of reward shaping as it easily distorts the actual objective. Specifically, assigning the penalty term " P_{Center} " (Sec. III.D.) is only feasible for training on synthetic data as ground truth for the updraft center is hardly available in a real-world scenario. From my point of view, the authors should at least state that fact. Even more so, as the agent seems to be capable of learning to exploit an updraft when this penalty term is assigned (cf. Fig. 12)*

Response (minor comment 2.4): Indeed, reward shaping which penalizes according to the distance from the thermal center is possible only in simulation, where the center position is known. We added the following text:

“The thermal center is not part of the agent’s state. Assigning such penalty is only feasible for training in simulations, as ground truth for the updraft center is hardly available in a real-world scenario. Yet, training with this penalty may still be indirectly advantageous in these cases (see Discussion).” (Line 169 in the clean version, line 264 in the track-changes version).

To reflect the indirect potential contribution of such a penalty term for improving real-world systems, we added the following sentence in the Discussion:

“While using a reward that penalizes the agent based on its distance from the thermal center is applicable only in simulations and not during training in the real-world, such a penalty may be of some value. Pre-training real-world agents in simulation with such a penalty may improve their ability to estimate the thermal position based on their available state information, thereby bridging some of the ‘sim-to-real’ gap”. (Line 507 in the clean version, line 604 in the track-changes version).

In Fig. 12 (current Fig. 4, orange lines in 4a and 4b) it is indeed shown that without the P_{center} penalty the agent managed to “survive” (*i.e.*, not crash, Fig. 4a), but it did not manage to stay close to the thermal center (Fig. 4b) and is, therefore, unable to exploit it for soaring.

Reviewer (concluding remarks 3): *Despite the criticism stated above, I want to point out that, after minor revision, I recommend publication. The technical content is sound. Moreover, the insights into the learning dynamics and trained agent's behavior are interesting. Notably, the display of information in Fig. 6, 7, 10, and Fig. 14 is excellent.*

Response: We are grateful for these comments!

Comments from Reviewer 3:

Reviewer: *This manuscript presents a study of thermal soaring in a simulated environment with horizontal winds by means of deep reinforcement learning techniques. The paper is well written and appears to be technically correct.*

Response: We appreciate the reviewer's opinion.

Reviewer: *However, the papers cited below have explored in depth this problem by a very similar approach, actually going beyond the content of this manuscript in at least two aspects*
1) *by using recurrent neural networks for memory based (and not just reactive) controls*
2) *testing in real world experiments*

In conclusion, I think this is a good paper but does not meet the high editorial threshold for broad impact and significance required for publication in Nature Communications.

*Notter, S., Schimpf, F., and Fichter, W., "Hierarchical Reinforcement Learning Approach Towards Autonomous Cross-Country Soaring," AIAA Scitech 2021 Forum, 2021
doi:10.2514/6.2021-2010.*

Notter, S., Schimpf, F., Müller, G., and Fichter, W., "Hierarchical Reinforcement Learning Approach for Autonomous Cross-Country Soaring," Journal of Guidance, Control, and Dynamics, Vol. 46, No. 1, 2023, pp. 114–126. doi:10.2514/1.G006746.

*Deep Reinforcement Learning Approach for Integrated Updraft Mapping and Exploitation
Stefan Notter, Christian Gall, Gregor Müller, Aamir Ahmad, and Walter Fichter
Journal of Guidance, Control, and Dynamics 2023 46:10, 1997-2004*

Also note this recent paper with a closely related approach

*Cui, Y.; Yan, D.; Wan, Z. Study on the Glider Soaring Strategy in Random Location Thermal Updraft via Reinforcement Learning. Aerospace 2023, 10, 834
<https://doi.org/10.3390/aerospace10100834>*

Response: We thank the reviewer for this insightful comment and for noting previous papers that we have missed to cite. Consequently, we entirely shifted the focus of the manuscript. Instead of focusing on the technical result of achieving thermal soaring using RL in high horizontal winds, the revised version highlights what we see as a broader perspective of using this system to address more basic questions related to the learning of motion control.

Indeed, previous works have achieved thermal soaring using RL in both simulation and real-world gliders, and in the revised manuscript we have cited all of the aforementioned works*. Recent works (Notter *et al.* 2023 and Cui *et al.* 2023) have used LSTM network architectures, which are, as the reviewers pointed, more sophisticated than the memory buffer architecture we have used, while both serve similar functionality. Hence, we leverage the recent successes in this field towards the context of using thermal soaring as a model system for studying the problem of learning motion control.

The highlights of the changes are listed above (response to comment 1.1 of Referee 2). Briefly, the changes are related to the bottleneck structure of the learning problem, the robustness of the RL solutions to sensor and environmental noise as well as to their own representation, pinning down the state and action variables crucial for high-wind soaring, comparing the RL solution with data from real vulture soaring (including additional analysis), and obtaining new insight into the inner workings of the agent's neural network in terms of its functional clusters. As such, we believe that the analyses and conclusions in the massively-revised manuscript are of interest to a wide audience of biologists, computer scientists, and engineers.

* The paper by Notter *et al.* 2021 appears to be an earlier version of Notter *et al.* 2023, hence the latter is cited in the manuscript.

REVIEWERS' COMMENTS

Reviewer #2 (Remarks to the Author):

The revised manuscript substantially improved over the initial submission. In particular, the authors shifted the focus to “study[ing] the learning process of thermal soaring.”

Minor technical comment: I would suggest to not refer to bank angle, flight path angle and heading angle as “roll”, “pitch” and “yaw”, in section 4.B. Roll, pitch and yaw commonly denote the aircraft attitude (in a 6-DoF context) and should not be mixed up with direction of flight (i.e., γ & χ) or the lift force (i.e., σ).

That said, the authors addressed all the issues raised in my initial review. Apart from the minor comment stated above, I recommend publication of the paper in its current form.

Reviewer #3 (Remarks to the Author):

The revised version is a significant improvement over the submitted one. I recommend publication.

Response to Referees

Manuscript: Revealing principles of autonomous thermal soaring in windy conditions using vulture-inspired deep reinforcement-learning

Overview: Reviewer 2 had a minor comment regarding the terminology of the glider's angles. This comment has been addressed. Reviewer 3 had no comments.

Comment from Reviewer 2:

Reviewer: *The revised manuscript substantially improved over the initial submission. In particular, the authors shifted the focus to “study[ing] the learning process of thermal soaring”.*

Minor technical comment: I would suggest to not refer to bank angle, flight path angle and heading angle as “roll”, “pitch” and “yaw”, in section 4.B. Roll, pitch and yaw commonly denote the aircraft attitude (in a 6-DoF context) and should not be mixed up with direction of flight (i.e., γ & χ) or the lift force (i.e., σ).

That said, the authors addressed all the issues raised in my initial review. Apart from the minor comment stated above, I recommend publication of the paper in its current form.

Response: The reference to the glider's bank angle, flight path angle and heading angle as roll/pitch/yaw (respectively) has been removed from the manuscript.